# Development of Novel Lipid-Based Formulations for Water-Soluble Vitamin C versus Fat-Soluble Vitamin D3

**DOI:** 10.3390/bioengineering9120819

**Published:** 2022-12-19

**Authors:** Jie Chen, Leila Dehabadi, Yuan-Chun Ma, Lee D. Wilson

**Affiliations:** 1Dr. Ma’s Laboratories Inc., Unit 4, 8118 North Fraser Way, Burnaby, BC V5J 0E5, Canada; 2Department of Chemistry, University of Saskatchewan, 110 Science Place, Saskatoon, SK S7N 5C9, Canada

**Keywords:** liposomes, vitamins, formulation, homogeneous systems, hydrophile–lipophile balance

## Abstract

The aim of this study was to develop a facile and novel lipid-based formulation of vitamin C and vitamin D3. Liposomes loaded with vitamin C and D3 were characterized using transmission electron microscopy (TEM) and zeta potential measurements for evaluating morphology, particle size and physical stability. HPLC was employed to quantify the content of vitamin C and vitamin D3 in their liposomal forms. The UHPLC analysis of the lipid-based vitamin formulation is an easy and rapid method for the characterization as well as the quantification of all components. In addition, encapsulation efficiency, vitamin loading and stability analysis were performed by the UHPLC method, in order to evaluate the reliability of the optimized lipid-based formulation. The TEM results provided key support for the core type of liposome structure in the formulations, whereas the HPLC results indicated that the liposomal vitamin C and D3 systems were homogeneous, and did not undergo phase separation. Taken together, the results demonstrate that liposomal encapsulated vitamins (vitamin C and D3) possess a unilamellar vesicle morphology with uniform particle size, despite differences in the hydrophile–lipophile profiles of the vitamins. The highly efficient encapsulation properties of such liposomal constructs are proposed to contribute to enhanced vitamin bioavailability.

## 1. Introduction

Contemporary developments in the pharmaceutical industry have seen an increased utilization of liposome carrier systems for the delivery of diverse compounds to target cells or to address drug toxicity [1,2,3]. Liposomes can simultaneously entrap both hydrophobic and hydrophilic substances, such as antimicrobials, antioxidants, flavor compounds, and bioactive constituents: this approach may serve to avoid decomposition prior to release of entrapped target compounds at the desired destination site [4,5,6]. Liposomes have drawn much attention from research and technological applications, due to their biocompatibility, biodegradability and low toxicity, including their capacity to capture both hydrophilic and lipophilic compounds [7,8]. Liposome assemblies can be achieved by amphiphilic lipids that self-assemble into bilayers driven by hydrophobic effects, which contribute to the unique structure function properties of liposomes as biomembranes and carriers for a variety of bioactive substances, due to their size, amphiphilic nature and biocompatibility [9]. Conventional liposomes are commonly classified into five types: multilamellar vesicles (MLV); small unilamellar vesicles (SUV); large unilamellar vesicles (LUV); multivesicular vesicles (MVV); and giant unilamellar liposomes (GUV) [10,11]. The size and homogeneity of the liposomes are the key properties to be considered for drug encapsulation, rather than the number of lamellar structures within the vesicles [9]. The ideal size of liposomes for drug delivery ranges between 50 and 200 nm [12,13]. Traditionally, liposomes may be prepared using a variety of techniques: reverse-phase evaporation; ether/ethanol injection; controlled hydration; electroformation; microfluidic channels; thin-film hydration (Bangham technique); detergent depletion; heating, etc. [14,15,16,17,18,19,20,21]. On the other hand, innovative technologies such as freeze-drying double emulsions, dual asymmetric centrifugation (DAC), and supercritical fluid (SCF) treatments have been developed, in the past decade, for a liposomal-dependent drug delivery system [22]. However, a number of these methods have shortcomings, such as complex processes, a potential denaturing risk for active compounds, technically challenging operating conditions, and low drug encapsulation efficiency (EE) [23]. Moreover, the use of harmful organic solvents poses disadvantages when manufacturing dietary supplements or therapeutic medicines. For the latter case, organic solvents must be excluded from liposome-based medicinal formulations, in order to meet food and drug safety standards, and thereby conform with the compliance and regulatory requirements of the pharmaceutical industry. As a result, production costs rise, due to the additional purification processes and hazardous material management procedures that are mandated by law [24]. Notably, several emerging trends in promising liposomal drug delivery methods have been recently reviewed, including stealth liposome technology, non-PEGylated liposome techniques, lysolipid thermally sensitive liposome techniques, and depo-foam liposome techniques [25]. 

To analyze liposomes, a detailed and comprehensive comparison of existing imaging methods has been reported, which includes techniques such as scanning electron microscopy (SEM), transmission electron microscopy (TEM), SEM with energy dispersive X-ray analysis (SEM–EDX), and atomic force microscopy (AFM). These techniques have been widely adopted for evaluating typical properties of nanoparticles, such as size, homogeneity, and zeta potential [26,27]. The production of liposomes in the pharmaceutical industry necessitates the rigorous quality control of both chemical composition and physicochemical properties: this requires sophisticated analytical methods, such as high-performance liquid chromatography (HPLC), dynamic light scattering (DLS), and transmission electron cryomicroscopy, to provide assurance of the content and stability of the chemical components, including the structure of the nanoparticles [28,29]. 

In industry, numerous food and nutraceutical companies have aspired to manufacture natural health products (NHPs) with high nutritional content, through vitamin fortification, in order to enhance pharmaceutical properties and provide therapeutic effects. Vitamin C is a water-soluble vitamin, among the most commonly adopted essential nutrients and bioactive compounds, which is added to various dietary supplements, due to its potential health benefits [30,31,32,33]. As a crucial reductant and antioxidant constituent acquired from foods and health care products, vitamin C is involved in a variety of physiological functions, including the prevention of molecular oxidation, the inhibition of the enzymatic browning process, oxygen scavenging to prevent oxidative deterioration, and the suppression of nitrosamine formation. Due to vitamin C’s high reactivity and sensitivity to oxidation, its degradation depends on various environmental factors, including temperature, pH, light, and oxygen, especially in the aqueous environments that occur in food and natural health products; therefore, an ongoing challenge is to maintain the stability of vitamin C during processing and preparation [30,31,32,33]. 

The major mechanism in aqueous media for vitamin C disintegration is oxidation to dehydroascorbic acid, which then rapidly converts to 2,3-diketogulonic acid, resulting in complete loss of the functional vitamin properties [34]. Although infusion methods via arteries or veins have higher bioavailability of vitamin C than oral administration, the disadvantages are obvious, such as strict administration criteria, risks of pathogen transmission, discomfort, and phlebitis [35,36,37]. The oral administration of vitamin C in the form of a crystalline powder or a liquid solution makes it prone to disintegration in the digestive system, particularly in the presence of metal ions; therefore, it is necessary to retard the breakdown of vitamin C in the stomach, and to enhance its absorption in the body, which can be achieved by encapsulation within the amphiphilic interior of the liposomes [38,39,40,41]. In addition to overcoming the oral bioavailability challenges and therapeutic issues of vitamin C, significant developments in drug optimization can improve orally administered health products, so that they reach the bloodstream by oral administration [42].

Various approaches have been employed, including microencapsulation, implementation of liposomes, and nanoparticles [43,44,45]. Lipid compounds that are usually regarded as harmless and biodegradable can promote transcellular delivery, by temporarily disrupting cellular lipophilic bilayers, as well as enhancing paracellular uptake of medicinal ingredients. Liposomal formulations of vitamin C not only delay its distribution, but can also increase absorption and prevent disintegration in the gut [38,39]. Generally, a healthy diet should contain lipids as a crucial element, especially phosphatidylcholines, as they have been shown to have a conducive impact on the overall wellness of patients [28,46,47]; in addition, the removal of organic solvents from the manufacturing process makes it challenging to achieve an ideal liposomal formulation; furthermore, the physical and chemical properties of phospholipids can change, upon the hydrolysis of ester linkages (between fatty acids and glycerol), the peroxidation of unsaturated acyl chains, and phospholipid degradation and oxidation, which may affect the quality and stability of the resulting liposomal forms [48,49]. 

Vitamin D is another important essential nutrient within a class of anti-rachitic compounds comprised of cyclopentanoperhydrophenanthrene rings, which are similar to those found in other steroids like cholesterol. In contrast to cholesterol, vitamin D is a three-ring system, and the structure of the side chains can distinguish naturally existing forms of vitamin D [50]. There are two main variants of vitamin D: vitamin D3 (active 7-dehydrocholesterol or cholecalciferol)—which is more commonly used, and is primarily transformed in dermal cells—and vitamin D2 (active ergosterol or ergocholecalciferol), which is mainly converted by irradiating ergosterol with UV light [51]. Specifically, the D3 form, or cholecalciferol, is a fat-soluble micronutrient that is required for several metabolic and immune processes. Vitamin D3 regulates about 700 genes, and consequently can control the in vivo metabolism of calcium, so as to increase intestinal calcium absorption, regulate phosphorus balance against osteoporosis, and thus promote normal bone formation and mineralization. Therefore, vitamin D3 deficiency may result in a variety of disorders, including bone disease, several cancers, type 1 diabetes, cardiovascular disease, and multiple sclerosis, especially in adults [52,53,54]. Humans can obtain vitamin D3 either intrinsically or extrinsically; however, an intrinsic source is only available by exposure to UVB, mostly from the sun, which raises the chance of melanoma. Based on such risks, ingestion of vitamin D3 is preferable, as the requirement of vitamin D3 from dietary intake is not typically met by the majority of accessible foods [55]. As vitamin D3 is particularly hydrophobic in nature, it is difficult to transport efficiently to the cells within the body; a change in composition, including a daily dosage of nutritional supplement, has a minor effect on the transportation of vitamin D3 and its derivatives throughout the body. Changes in vitamin formulation may have a significant impact on digestion and subsequent absorption mechanisms [56], as the biopolymer matrix found in mucus is composed of mucins, which inhibit digestion, followed by the absorption of particles containing hydrophobic substances, where particles less than 300–500 nm may only reach the epithelial cells [57]. Moreover, only competent cells may uptake the entire particle cargo of specific sizes, which allows for the endocytosis of highly hydrophobic molecules [58,59]. These features, along with the knowledge of physiologically relevant particles, have led to the design of an efficient formulation for highly hydrophobic substances in the context of liposomes, which are widely used as carriers to prevent degradation and increase the shelf life of active medicinal ingredients [60]. In addition, the low solubility of vitamin D3 prevents homogeneous physiological distribution, where liposomal vitamin D3 may further enhance the solubility profile, and regulate the release of medicinal compounds [61]. Herein, we demonstrate a robust lipid-based formulation for vitamin C and vitamin D, which is homogenous and stable without the requirement of any harmful chemicals or solvents in the formulation process.

## 2. Materials and Methods

Vitamin C (99.66% sodium ascorbate) and vitamin D3 (98.50% cholecalciferol) were purchased from Sigma-Aldrich (New York, NY, USA). In addition, sunflower lecithin (≥40% phosphatidylcholine) was obtained from Jedwards International, Inc. (Braintree, Chicago, IL, USA). Potassium sorbate (PS) (99.47 ± 0.5%) and glycerin (99.80%) were purchased from Quadra Chemicals (Delta, BC, Canada) and EZ Chemicals Inc. (Mississauga, ON, Canada), respectively. Olive oil (100.00%) (Kirkland, WA, USA) and distilled water (ELGA LabWater (Woodridge, IL, USA)) were used for the preparation of all solutions. Methanol (99.80% HPLC grade) and acetonitrile (99.99% HPLC grade) were purchased from VWR international, LLC. Phosphoric acid (85.00% HPLC grade) was obtained from EMD Millipore Corporation (Kankakee, IL, USA). All solvents and samples were filtered through 0.2 μm wwPTFE filter (Pall Corporation (New York, NY, USA)) before injection into the HPLC.

### 2.1. Preparation of Liposomal Vitamin C

A solution was prepared at room temperature (25 °C), by adding 0.134 g of potassium sorbate (PS) (99.47 ± 0.5%), and dissolving in purified water (45 mL). Then, 1124 (+5%) mg of vitamin C (99.66% sodium ascorbate) was added to the aqueous solution, and stirred until completely dissolved. There was 5% overage of material input for vitamin C, according to the sample product formula. Next, 55 mL glycerin (99.80%) was added to an aqueous solution, by stirring; the solution was then placed in a blender (Hamilton Beach (Avalon, NJ, USA)), and was mixed thoroughly at a high speed (3000 rpm). Then, sunflower lecithin (≥40% PC) (1.66 g) was added to the aqueous solution, and was blended at 3000 rpm to obtain liposome encapsulated vitamin C (cf. Figure 1a).

### 2.2. Preparation of Liposomal Vitamin D3

A hydrophilic solution was prepared at room temperature (25 °C), by weighing 0.134 g of PS (99.47 ± 0.5%), and dissolving this amount in 45 mL of purified water. Next, 55 mL of glycerin (99.80%) was added to the aqueous solution, by stirring; the solution was then placed in a blender container (Hamilton Beach (Avalon, NJ, USA)), and was thoroughly mixed at high speed (3000 rpm). Then, 1.25 (+10%) mg of vitamin D3 (98.50% cholecalciferol) was dissolved in 0.3 mL of olive oil (100.00%), which was added to the mixture, and blended at 3000 rpm for 30 min. There was 10% overage of material input for vitamin D3, according to the sample product formula. Next, 1.66 g of sunflower lecithin (≥40% phosphatidylcholine; PC) was added to the above solution, followed by blending, to obtain liposome-encapsulated vitamin D3 (cf. Figure 1b).

### 2.3. Characterization of Vitamin C and Vitamin D3 in Liposome Forms

#### 2.3.1. Transmission Electron Microscopy (TEM) Analysis

TEM analysis was performed by negative staining. Briefly, 5 µL of liposomal sample was placed on a copper–formvar-coated TEM grid, and was allowed to settle on the grid surface for 2 min. The excess liquid was removed using an absorbent tissue. Staining of the grid was done using 0.5% phosphotungstic acid for 20 s, and excess staining was removed. Imaging was achieved using an HT 7700 TEM (Tokyo, Japan) at 80 kV.

#### 2.3.2. Particle Size Distribution and Zeta Potential 

The particle size and zeta potential of the liposomes were measured using a Malvern Zetasizer Nano ZS instrument (Malvern Instruments Ltd., Malvern, UK). The sample size distribution was obtained by measurement of scattered light (θ = 173°) by particles (dynamic light scattering, DLS) illuminated with a laser beam. Zeta potential was established by an electrophoretic light scattering method based on the Doppler effect [62]. Zeta potential charge values were derived from triplicate measurements, where each consisted of a minimum of ten individual runs.

#### 2.3.3. Encapsulation Efficiency (*EE*%) and Vitamins Loading (*VL*%)

The encapsulation efficiency (*EE*%) of the vitamins in the liposomes was calculated by Equation (1) [63,64,65]:(1)$$EE\;\left(\%\right)=\frac{Total\;vitamin\;content\;\left(\mathrm{mg}\right)-Solution\;vitamin\;content\;\left(\mathrm{mg}\right)}{Total\;vitamin\;content\;\left(\mathrm{mg}\right)}\times 100$$

Nonencapsulated vitamins in the solution were quantified by high-performance liquid chromatography, after ultrafiltration (10 kDa) and centrifugation (Nuaire, NU-C200R, Plymouth, MA, USA) at 2500× *g* for 20 min at 4 °C [63]. The absorbance demonstrated the nonencapsulated vitamin content; thus, the amount of trapped vitamin was calculated indirectly by Equation (1).

Vitamin loading (*VL*%) means the amount of vitamin that has been encapsulated in hydrated liposomes, and it was obtained by employing Equation (2) [64,65,66]:(2)$$VL\;\left(\%\right)=\frac{Vitamin\;content\;in\;liposomes\;\left(\mathrm{mg}\right)}{Weight\;of\;liposomes\;\left(\mathrm{mg}\right)}\times 100$$

#### 2.3.4. Instrumentation and Chromatographic Conditions

The UHPLC system consisted of flexible pumps, multicolumn thermostats, vial samplers, and diode array detector (Agilent 1290 Infinity II (Santa Clara, CA, USA)). The separation of vitamin C was carried out with a gradient elution program at a flow rate of 0.5 mL min^−1^ at 30.0 °C, by using a Luna C18 (100 mm × 3.0 mm, 2.5 μm) column supplied by Phenomenex (Torrance, CA, USA). The injection volume in the UHPLC system was 1 μL, where the mobile phase consisted of 0.05% (*v*/*v*) phosphoric acid (A) and methanol (B). The following linear gradient was applied: 0–3 min: 100% A; 3–5 min: 100–0% A, followed by re-equilibration of the column for 5 min before the next run. A wavelength of 245 nm was used for analyte detection of vitamin C. The separation of vitamin D3 was carried out with an isocratic elution program at a flow rate of 0.5 mL min^−1^ at 30.0 °C, by using a Luna C18 (100 mm × 3.0 mm, 2.5 μm) column supplied by Phenomenex, USA. The injection volume in the UHPLC system was 1 μL, and the mobile phase consisted of 75% acetonitrile (A) and 25% methanol (B). The total running time of the isocratic elution was 10 min. A wavelength of 280 nm was used for the detection of vitamin D3.

#### 2.3.5. Sample Preparation for UHPLC Analysis 

Samples of liposomal vitamin C from the top and bottom regions of the centrifuge vial were mixed with 50% methanol, at volume ratio 1:1000 (*v*/*v*), and were sonicated for 30 min before filtration and injection. A volume of 0.8 mL of liposomal vitamin D3 from the top and bottom portions of the sample tube was mixed with 100% methanol, and then the volume was brought to 10 mL in a volumetric flask. The sample was sonicated for 30 min before filtration and injection.

#### 2.3.6. Liposomal Stability 

The stability of the liposomes was assessed by determining the remainder of the vitamin in the liposomes after 3 months of storage at a refrigerated temperature of 4 °C, using Equation (3) below [63]:(3)$$Stability\;\left(\%\right)=\frac{Remaining\;amount\;of\;vitamin\;in\;liposomes\;\left(\mathrm{mg}\right)}{Initial\;amount\;vitamin\;incorporated\;in\;liposomes\;\left(\mathrm{mg}\right)}\times 100$$

## 3. Results and Discussion

As outlined above, a key goal of this study was to develop a robust lipid-based formulation for vitamin C and vitamin D, where the resulting system is homogenous and stable without requiring any harmful chemicals or solvents. This section describes the characterization of the lipid-based formulations that contain vitamin C and D, according to a range of methods (TEM imaging, light scattering, and HPLC). Various parameters—including particle size, zeta potential, encapsulation efficiency, and vitamin loading—which provide a measure of the stability of liposomal formulations, are discussed in the following sections. 

### 3.1. Optimization of Formulation

Based on previous research [67], this study developed an optimized lipid-based formulation, to achieve high stability and homogeneity, where the liquid composition was critical— especially the ratio of water to glycerin, which played a key role in the sample preparation. By comparison of samples with different water–glycerin ratios, it was determined that a 45:55 (*v*/*v*) ratio should be used as the optimized water–glycerin volume ratio. For the conditions in which glycerin had a higher volume than water, the liposomes could be formed after homogenization, where the order of mixing of the respective ingredients also had a dramatic impact on the uniformity of the liposome product.

### 3.2. Morphology by TEM 

The TEM imaging results in Figure 1 show that the liposomes were spherical in size, and that small fragments of liposomes were highly dispersed throughout them, which may indicate vesicle fracture [68,69]. Image J software (version 1.52, National Institutes of Health, Bethesda, MD, USA) was used to estimate the mean particle size of the liposomes, which was anticipated to be near 200 nm for liposomal vitamin C, and near 100 nm for liposomal vitamin D3. It is generally recognized that the size of liposomes has a significant role on tissue cell targeting: larger liposomes are usually taken by phagocytes, whereas smaller liposomes (100 to 200 nm) are more permeable to tumor cells [66,70]. The TEM images of liposomal vitamin C and D3 indicate that the liposomes were bilayer (unilamellar) vesicles, where the average sizes ranged between 100 nm to 200 nm.

### 3.3. Size and Zeta Potential

The surface charge, electrostatic repulsion between neighboring particles, colloidal stability, and entrapment efficiency were correlated with the zeta potential (ζ) [71]. According to the ζ-value results, the liposomes that contained vitamin C had a lower ζ-value (0.8 mV) than the pristine vitamin C solution [72], which related to the enhancement effect of the carrier between the phospholipids and the active compounds. The stability of the vitamin D3 liposomes could also be characterized by the ζ-value. Based on the result, a negative ζ-value (−4.0 mV) for liposomal vitamin D3 indicated stronger particle repulsion; therefore, there was no driving force for the particles to associate with, resulting in a more stable and dispersed liposome without coagulation [63,73,74]. The PDI values were close in magnitude, and exceeded 0.2 for liposomal vitamin C (0.2) and liposomal vitamin D3 (0.53), which indicated monodisperse and polydisperse particle size distributions, respectively [75]. Liposomes may vary in size, between 200–3000 nm for liposomal vitamin C, and 100–1000 nm for liposomal vitamin D3, which is related to multiple lipid layers, fusion or aggregation phenomena [75,76].

### 3.4. Evaluation of Encapsulation Efficiency (EE%) and Vitamin Loading (VL%) 

Table 1 summarizes the encapsulation efficiency and vitamin loading of liposomal vitamin C and liposomal vitamin D3. These parameters depended mostly on the composition of the liposomes, and may have been affected by other factors, such as liposome size and type, surface charge, bilayer rigidity, preparation method, and the properties of the vitamins. In turn, such factors may have helped to enhance the stability of the liposomes, and to have enhanced bioavailability on a physiological level without sacrificing its potency on the cellular level in the digestive system [77,78].

### 3.5. Determination of Vitamin C Potency in Liposomes 

The potency of ascorbic acid in the lipid-based formulation of vitamin C was measured by UHPLC equipped with a DAD detector, at a wavelength of 245 nm. Table 2 indicates the concentration of vitamin C from the top and bottom regions of the sample tube (cf. inset in Figure 2a,b); the vitamin recovery values were 97.38% and 91.86%, respectively. Based on these results, the sample was homogenous, and did not show any significant phase separation; however, the concentration of the top portion was greater than the bottom region. According to Health Canada’s regulation of Natural Health Products, a variation of 20% in composition is allowed for industrial production of such vitamin formulations. 

### 3.6. Determination of Vitamin D3 Potency in Liposomes 

The potency of cholecalciferol in liposomal vitamin D3 was assessed by UHPLC equipped with a DAD detector, at a wavelength of 280 nm. Table 3 shows the concentration of vitamin D3 determined from the top and bottom regions of the sample tube (cf. insets in Figure 3a,b); the vitamin recovery values were 100.95% and 96.73%, respectively. Based on these results, the sample was homogenous, and did not display any notable phase separation. According to Health Canada’s regulation of Natural Health Products, a variation of 20% in composition is allowed for industrial production of such vitamin formulations. 

### 3.7. Comparative Analysis of Vitamin Formulations

Size is a critical factor in determining the stability, biological origin, and effectiveness of formulated bioactives [79,80]. In this comparison study of liposomal vitamin C and liposomal vitamin D3, the particle size of liposomal vitamin C (~200 nm) was greater than that estimated for liposomal vitamin D3 (~100 nm), due to the higher bioactive–lipid weight ratio (B/L ratio), suggesting that the concentration of vitamin C was much greater than that of vitamin D3 [81]. While a 1:1 ratio is advantageous from a commercial standpoint, less phosphatidyl choline is required for its formulation. A 1:2 ratio, by comparison, achieves a smaller particle size (below 200 nm), which is optimal for liposomal stability: the latter is an ideal vesicle size (less than 200 nm) that is consistent with various food-based liposomes [82,83,84]. As a result, the upper region has a concentration of liposomal vitamin C that is higher than the bottom region of the sample. This trend indicates less homogeneity, unlike liposomal vitamin D3, where the concentration near the top region was very similar to the bottom region of the sample, indicating that a more uniform and stable solution was achieved with vitamin D3. This comparison of results from different sampling regions is consistent with the established relationship of the particle size and B/L ratio, which consequently leads to its stability [81]. We have further confirmed that at higher B/L ratios, colloids with larger size will be generated. Liposomes with reduced homogeneity concur with liposome systems that display reduced stability of the lipid-based formulation.

### 3.8. Stability 

The results of the present study, shown in Table 4 and Table 5, reveal that the liposomal entrapped vitamins leaked out only slightly over a 3-month period, for both 4 and 25 °C, as compared with the samples that were assessed immediately after the liposomal preparation. 

No significant differences in vitamin concentration were found between the top and bottom, for intervals ranging from 0 days to 90 days. This trend indicates that the formulations of both lipid-based vitamin C and vitamin D3 are homogenous, and stable for a minimum time of 3 months.

## 4. Conclusions

HPLC has been widely used for the identification and quantification of vitamins in diverse matrices, due to its relatively rapid separation capability, high sensitivity, precise quantification, and effective analytical performance; therefore, HPLC was chosen as the ideal method for assessing the potency of vitamin C and D3 in liposomal forms, and as a suitable analytical method for regular laboratory testing in the pharmaceutical industry. Based on a previous study [66], the encapsulation efficiencies of vitamin C were generally around 50%, which compares favorably to our results. A comparison of hydrophilic vitamin C to lipophilic vitamin D3 revealed a higher encapsulation efficiency (>90%) for vitamin D3, due to its high lipophilicity versus the more water-soluble (hydrophilic) vitamin C [85]. This trend was consistent with similar studies, and is related to the decreased particle size and polydispersity index, due to the role of the relative proportion of the constituents, the viscosity of the lipid phase, and the production conditions [86,87].

Our results provide support for both types of liposomal vitamin C and D3, which are homogenous colloids that remain very stable without phase separation, for a minimum of 3 months. Lipid-based formulations of this type are considered to be an effective and versatile approach to addressing the fortification of nutraceuticals and other bioactive components with variable molecular properties that cover a range of values on the hydrophile–lipophile scale. Our results concur with those of a recent study of liposomal-based vitamin C that had been readily prepared with food grade materials and simple equipment, and which provided a well-tolerated natural health product. The uptake of ingested liposomes from the gut to the mesenterial vessels and liver vessels was supported by a microbubble-enhanced ultrasound (MEU) method with real-time imaging [76]. Similarly, liposomes with a well-defined size distribution were used to achieve efficient absorption of hydrophobic vitamin D3. Clinical trials indicate that orally delivered liposomal vitamin D3 enhances the concentration of calcidiol in the serum rapidly, whereas no such function is found in oil-based vitamin D3 liquid [88].

This work focused mainly on the optimization of a lipid-based formula of water-soluble and fat-soluble vitamins, using a simple method and food-grade materials. Further studies could be extended to other applications of liposomal assemblies: for example, skin permeation studies could be carried out for both lipid-based vitamin C and lipid-based vitamin D3, in order to evaluate liposomes cellular internalization, and also to provide insight regarding the behavior of nanoparticles upon interaction with biological membranes [66,89,90]. In the case of nanocarriers, these systems have advantages for the topical and transdermal application of vitamin D3 (active 7-dehydrocholesterol or cholecalciferol), which is transformed primarily in dermal cells [51,66].

## Data Availability

Not applicable.

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
