# Peer review of "Development of Novel Lipid-Based Formulations for Water-Soluble Vitamin C versus Fat-Soluble Vitamin D3"

_bioengineering, 2022, doi:10.3390/bioengineering9120819_

Round 1

Reviewer 1 Report

This manuscript focuses on the development of a robust lipid-based formulation for vitamin C and vitamin D. This formulation is homogenous and stable without the requirement of any harmful chemicals or solvents in the formulation process. Antioxidant liposomal systems attract attention by their great potential for cosmetic, food, and nanomedicine applications. The work needs major revision with an additional critical experiment.

Major comments

1. Article title is rather broad. It is better to be specific, because this paper is predominantly about formulations only for two (C, D) vitamins.

2. It is better to use term “the lipid-based formulations” in all text of article. Liposomal formulations need more analytical methods for characterization.

3. The introduction is very long. It is better to shorten and keep important points related to the goals of this work.

4. Page 1, line 40-44 “Traditionally liposomes may be prepared using a variety of techniques: reverse-phase evaporation, ether/ethanol injection, controlled hydration, electro formation, microfluidic channel, thin-film hydration (Bangham technique), membrane extrusion, detergent depletion, heating, homogenization, and sonication, etc.”

Please, correct because techniques like membrane extrusion, sonication, homogenization and/or freeze-thawing are being employed to control the size and size distribution…

5. Page 4 2. Materials and Methods. Please indicate the purity of all reagents (vitamins, lecithin and etc.)

6. Page 4, line 169 2.1. Preparation of Liposomal Vitamin D3 and page 4, line 177 2.2. Preparation of liposomal Vitamin C

What is the method used for preparation of liposomes?

page 4, line 173 and page 4 line 182 “mixed at high speed thoroughly” what was the speed? It is better to indicate the temperature of liposomal preparation at each step. What is the sunflower lecithin phase-transition temperature?

7. page 4, line 185-186 2.3.1. Transmission Electron Microscopy (TEM) “a drop of liposomal sample” the concentration is needed.

8. Page 4, line 191-196 2.3.2. Particle size Distribution and Zeta-Potential. Angle measurement is needed.

9. Page 5, line 219 3.1. Optimization of formulation. “this study developed an optimized liposomal formulation to achieve high stability and homogeneity, where the liquid composition is critical, especially the ratio of water and glycerin plays a key role in sample preparation” Which volumes were optimized? Only two volumes were indicated: the amount in 45 ml purified water (page 4, line 179) and 55 ml glycerin (page 4, line 180).

10. Page 5, fig. 1. The size and shape of lipid nanoparticles are different on the fig. 1A. What was the polydispersity index? How it was changed at optimization liposomal formulation step? How was the stability?

11. Page 5, line 237 3.3. Size and zeta potential. What were the size, PDI and stability? Zeta is close to zero, which indicates possible low stability of nanoparticles.

12. Vitamin loading calculation and encapsulation efficiency are needed to characterize the liposomal formulations of vitamins C and D.

13. Are vitamins released from lipid systems?

14. Conclusion, page 8, line 82 “which are homogenous and remain stable without phase separation remain stable without phase separation” How long?

Author Response

Authors Response to Reviewer comments on MS ID:  bioengineering-2047251

Reviewer #1

This manuscript focuses on the development of a robust lipid-based formulation for vitamin C and vitamin D. This formulation is homogenous and stable without the requirement of any harmful chemicals or solvents in the formulation process. Antioxidant liposomal systems attract attention by their great potential for cosmetic, food, and nanomedicine applications. The work needs major revision with an additional critical experiment.

Major comments

  1. Article title is rather broad. It is better to be specific, because this paper is predominantly about formulations only for two (C, D) vitamins.

 Response: Thank you for your suggestion. The title has been revised to “Development of Novel Lipid-based Formulations for Water-soluble Vitamin C versus Fat-soluble Vitamin D3”.

  1. It is better to use term “the lipid-based formulations” in all text of article. Liposomal formulations need more analytical methods for characterization.

 Response: The authors wish to thank the reviewer for this important comment. The terms of “the lipid-based formulation” has been applied on all text of article. Encapsulation efficiency and vitamin loading for lipid-based vitamin C and lipid-based vitamin D3 have been added to the revised manuscript as part of liposome characterization techniques.

  1. The introduction is very long. It is better to shorten and keep important points related to the goals of this work.

 Response: We revised the introduction to remove the unrelated part and only address the main goals of this work.

  1. Page 1, line 40-44 “Traditionally liposomes may be prepared using a variety of techniques: reverse-phase evaporation, ether/ethanol injection, controlled hydration, electro formation, microfluidic channel, thin-film hydration (Bangham technique), membrane extrusion, detergent depletion, heating, homogenization, and sonication, etc.”

Please, correct because techniques like membrane extrusion, sonication, homogenization and/or freeze-thawing are being employed to control the size and size distribution…

 Response: We agree with the reviewer and “Traditionally liposomes may be prepared using a variety of techniques: reverse-phase evaporation, ether/ethanol injection, controlled hydration, electro formation, microfluidic channel, thin-film hydration (Bangham technique), membrane extrusion, detergent depletion, heating, homogenization, and sonication, etc.” The current version has been revised to “Traditionally liposomes may be prepared using a variety of techniques: reverse-phase evaporation, ether/ethanol injection, controlled hydration, electro formation, microfluidic channel, thin-film hydration (Bangham technique), detergent depletion, and heating, etc.” 

  1. Page 4 2. Materials and Methods. Please indicate the purity of all reagents (vitamins, lecithin and etc.)

 Response: Thank you for the comment. Purity of all reagents was added to the revised manuscript.

  1. Page 4, line 169 2.1. Preparation of Liposomal Vitamin D3 and page 4, line 177 2.2. Preparation of liposomal Vitamin C

 Response: The preparation of lipid-based vitamins was updated according to reviewer’s comment.

What is the method used for preparation of liposomes?

 Response: The modified homogenization method was used for preparation of liposomes [1,2].

  1. Jeung, J.J. Vitamin C delivery system and liposomal composition thereof. US Patent 0367480 A1, filed 17 Jun 2016, and issued 22 Dec 2016.
  2. Poudel, A.; Gachumi, G.; Wasan, K.M.; Dallal Bashi, Z.; El-Aneed, A.; Badea, I. Development and Characterization of Liposomal Formulations Containing Phytosterols Extracted from Canola Oil Deodorizer Distillate along with Tocopherols as Food Additives. Pharmaceutics, 2019, 11(4), 185-201.

page 4, line 173 and page 4 line 182 “mixed at high speed thoroughly” what was the speed? It is better to indicate the temperature of liposomal preparation at each step. What is the sunflower lecithin phase-transition temperature?

 Response: The authors wish to thank review for this feedback. The speed (3000 rpm) and temperature (25℃) have been indicated in revised manuscript. The phase-transition temperature of sunflower lecithin is 37˚C [1].

  1. Kalmanovich, S.A.; Butina, E.A.; Gerasimenko, E.O.; Butina, E.A.; Kharchenko, S.A. The Use of Fractionated Sunflower Lecithins for Encapsulation of Micronutrients. Asian J. Pharm. 2016, 10(3), 386-393.
  2. page 4, line 185-186 2.3.1. Transmission Electron Microscopy (TEM) “a drop of liposomal sample” the concentration is needed.

 Response: We agree with reviewer and the concentration was added to the revised manuscript.

  1. Page 4, line 191-196 2.3.2. Particle size Distribution and Zeta-Potential. Angle measurement is needed.

  Response: The authors wish to thank the reviewer for this comment. The angle measurement has been added to the revised manuscript.

  1. Page 5, line 219 3.1. Optimization of formulation. “this study developed an optimized liposomal formulation to achieve high stability and homogeneity, where the liquid composition is critical, especially the ratio of water and glycerin plays a key role in sample preparation” Which volumes were optimized? Only two volumes were indicated: the amount in 45 ml purified water (page 4, line 179) and 55 ml glycerin (page 4, line 180).

 Response: The ratio of the volume of water and glycerin was optimized, where the ratio of water and glycerin is (1:1.2), which is very important to the formulation of liposomes. Different ratios were applied and only this particular ratio appears to form a homogenous and stable emulsified solution.

  1. Page 5, fig. 1. The size and shape of lipid nanoparticles are different on the fig. 1A. What was the polydispersity index? How it was changed at optimization liposomal formulation step? How was the stability?

 Response: Negative staining TEM is a common method for characterization of liposomes since it is simple, fast and requires little material with results of high contrast images. Negative staining of liposomes could have artefacts since lipids are not “fixed” in their native structure by uranyl stains [5, 6], and the interaction with the carbon film surface usually results in spreading of lipids and possible recombination of lipids to hydrophilic films or other structures. Large liposomes may rupture during the drying process or fold into irregular shapes. When liposome samples interact strongly with the carbon film, further dilution of the liposomes and variation of the protocol can avoid some of the issues. Thus, the results must be carefully analyzed and interpreted. Unexpected observations/structures (e.g., open lipid films, rippled lipid phases, worm-like micelles and tubular structures, bicelles, micelles, etc.) [1]. The optimized formulation was determined after a homogenous and stable solution was achieved without any observable phase separation. The polydispersity index and stability of the optimization liposomal formulation were added to the revised manuscript.

  1. Baxa, U. Imaging of Liposomes by Transmission Electron Microscopy. Methods in Molecular Biology, 2018, 1682, 73-88.
  2. Page 5, line 237 3.3. Size and zeta potential. What were the size, PDI and stability? Zeta is close to zero, which indicates possible low stability of nanoparticles.

 Response: The authors agree with reviewer and the size, PDI and stability comments, which were added to the revised manuscript. The zeta potential value is not the only sign of nanoparticle stability. The stability of the liposomes is related to the lengths and the saturation of the fatty acid chains. The more saturated fatty acid chains composing the bilayer, the assembled liposome structure becomes more stable. Zeta potential can be affected by many factors, such as the liposome composition, charged lipids, the pH, etc. [1].

  1. Németh, Z.; Csóka, I.; Semnani Jazani, R.; Sipos, B.; Haspel, H.; Kozma, G.; Kónya, Z.; Dobó, D. G. Quality by Design-Driven Zeta Potential Optimisation Study of Liposomes with Charge Imparting Membrane Additives. Pharmaceutics, 2022, 14, 1798-1823.

  1. Vitamin loading calculation and encapsulation efficiency are needed to characterize the liposomal formulations of vitamins C and D.

 Response: The authors wish to thank the reviewer for this comment. The Vitamin loading calculation and encapsulation efficiency were added to the revised manuscript.

  1. Are vitamins released from lipid systems?

 Response: For vitamin C, liposomal formulations are inferred to improve compound release in terms of being harmless and biodegradable, which are able to promote transcellular delivery by temporarily disrupting cellular lipophilic bilayers, including enhancement of paracellular uptake of medicinal ingredients. Liposomal formulations of vitamin C not only delays its distribution, it can also increase absorption and prevent disintegration in the gut [1,2].

For vitamin D3, the corresponding liposomal formulation improves compound release in three aspects: reduction or elimination of degradation, high absorption and homogenous distribution of vitamin D3 including particle size (particles less than 300-500 nm may only reach the epithelial cells [3]), cell type (only competent cells may uptake the entire particle cargo of specific sizes, which allow for the endocytosis of highly hydrophobic molecules [4,5]) and solubility (liposomal vitamin D3 may further enhance the solubility profile and regulate release of medicinal compounds [6]).

  1. Wechtersbach, L.; Poklar Ulrih, N.; Cigic, B. Liposomal stabilization of ascorbic acid in model systems and in food matrices. LWT- food sci. technol. 2012, 45(1), 43–49.
  2. Hickey, S.; Roberts, H.J.; Miller, N.J. Pharmacokinetics of oral vitamin C. Nutr. Environ. Med. 2008, 17(3), 169–177.
  3. Cone, R.A. Barrier properties of mucus. Drug Deliv. Rev. 2009, 61,75-85.
  4. Guo, Q.; Bellissimo, N.; Rousseau, D. The physical state of emulsified edible oil modulates its in vitro digestion. Agric. Food Chem. 2017, 65, 9120-9127.
  5. Gupta, R.; Behera, C.; Paudwal, G.; Rawat, N.; Baldi, A.; Gupta, P.N. Recent advances in formulation strategies for efficient delivery of vitamin D. AAPS Pharm. Sci. Tech. 2019, 20(1), 11-23.
  6. Guo, F.; Lin, M.; Gu, Y.; Zhao, X.; Hu, G. Preparation of PEG-modified proanthocyanidin liposome and its application in cosmetics. Food Res. Technol. 2015, 240, 1013–1021.

  1. Conclusion, page 8, line 82 “which are homogenous and remain stable without phase separation remain stable without phase separation” How long?

 Response: Based on the stability study for real time stability results, the lipid-based formula can be stable for 1 year of shelf life. The data was added to the revised manuscript.

In summary, the authors wish to acknowledge reviewer #1 for the insightful and constructive comments. The revised manuscript was comprehensively edited for language, clarity, and syntax throughout to meet the high standards of this journal.

Reviewer 2 Report

The current manuscript deals with the preparation of Vitamin C and D3 liposomal formulations.  The procedure to prepare this formulation is optimized and loading efficiency was compared. The authors used TEM to characterize the morphology of the liposome assemblies.  Even though the work presented here has several interesting features such as lipid formulations and their characterization, this work lack the application of the prepared liposomal assemblies. 

The authors gave appreciable introcution to lipid formulation, but they lack showing the application of the formulations such as their delivery or the stability for liposome complex.

I suggest the manuscript should ab accepted after a major revision which includes the study of the stability of the lipid formulation and their controlled release studies as mentioned in their introduction.

One more point to consider is that the introduction of the paper is too long and has one paragraph, which makes it harder to read. I suggest dividing the introduction into a few paragraphs for better readability.

Author Response

Authors Response to Reviewer comments on MS ID:  bioengineering-2047251

Reviewer #2

The current manuscript deals with the preparation of Vitamin C and D3 liposomal formulations.  The procedure to prepare this formulation is optimized and loading efficiency was compared. The authors used TEM to characterize the morphology of the liposome assemblies.  Even though the work presented here has several interesting features such as lipid formulations and their characterization, this work lack the application of the prepared liposomal assemblies. 

Response: Thank you for valuable comments. This work is mainly focused on optimization of the lipid-based formula of water soluble and fat-soluble vitamins that employs a simple method and food grade materials. Further investigation may be involved in application of liposomal assemblies.

For example, skin permeation studies can be carried out for both lipid-based vitamin C and lipid-based vitamin D3, in order to evaluate liposomes cellular internalization and provide explanations regarding the behavior of nanoparticles upon interaction with biological membranes [1-3]. In particular, nanocarriers have advantages in the topical and transdermal application of vitamin D3 (active 7-dehydrocholesterol or cholecalciferol) which is primarily transformed in dermal cells [4, 1].

  1. Maione-Silva, L.; De Castro, E.G.; Nascimento, T.L.; Cintra, E.R.; Moreira, L.C.; Cintra, B.A.S.; Valadares, M.C.; Lima, E.M. Ascorbic acid encapsulated into negatively charged liposomes exhibits increased skin permeation, retention and enhances collagen synthesis by fibroblasts. Rep. 2019, 9, 522-536.
  2. Bonnekoh, B.; Roding, J.; Krueger, G. R.; Ghyczy, M.; Mahrle, G. Increase of lipid fluidity and suppression of proliferation resulting from liposome uptake by human keratinocytes in vitro. BJD, 1991,124, 333–340.
  3. White, P.J.; Fogarty, R.D.; McKean,C.; Venables, D.J.; Werther, G.A.; Wraight, C.J. Oligonucleotide uptake in cultured keratinocytes: influence of confluence, cationic liposomes, and keratinocyte cell type. JID, 1999, 112, 699–705.
  4. Bender, D.A. Nutritional Biochemistry of the Vitamins. Cambridge University Press: New York, 2003.

The authors gave appreciable introduction to lipid formulation, but they lack showing the application of the formulations such as their delivery or the stability for liposome complex.

Response: The authors wish to thank the reviewer for this comment. The introduction was revised according to the main goal of this work, which is focused on optimization of the lipid-based formula of water soluble and fat soluble vitamins that employs a simple method with food grade materials. Further investigation may be involved in application of liposomal assemblies.

I suggest the manuscript should ab accepted after a major revision which includes the study of the stability of the lipid formulation and their controlled release studies as mentioned in their introduction.

Response: The stability results of lipid-based formulation were added to the revised manuscript. The controlled release studies [1-3] are related to application of liposomal formulation and may be considered in future investigation.

  1. Blesso, C.N. Egg phospholipids and cardiovascular health. Nutrients, 2015, 7(4), 2731–2747.
  2. Kullenberg, D.; Taylor, L.A.; Schneider, M.; Massing, U. Health effects of dietary phospholipids. Lipids Health Dis. 2012, 11(3), 1-16.
  3. Garcia, J.T.; Aguero, S.D. Phospholipids: properties and health effects. Hosp. 2015, 31(1), 76–83.

One more point to consider is that the introduction of the paper is too long and has one paragraph, which makes it harder to read. I suggest dividing the introduction into a few paragraphs for better readability.

Response: The authors wish to thank the reviewer for this comment. The introduction was updated in the revised manuscript.

In summary, the authors wish to acknowledge reviewer #2 for the insightful and constructive comments. The revised manuscript was comprehensively edited for language, clarity, and syntax throughout to meet the high standards of this journal.

Reviewer 3 Report

The manuscript is devoted to the study of novel formulation of liposomal vitamin C and vitamin D3. I believe that this manuscript is interesting and should be publishable in this journal; however there are several scientific aspects of this manuscript that I feel the authors must first address.

1. The abstract must be rewritten and will be contain more detail experimental information.

2. line 35-37. Authors indicated on the four different type of liposome. Please, clarify the absence of the information about of giant unilamellar liposomes (GUV).

3. line 59-60. Please, explain the abbreviations.

4. The introduction a very large collection of different information, and refers to many other findings in the literature. A clear message is missing as the paper is very long and poorly structured.

5. Please, clarify the choice of the lipids composition of the novel formulation.

6. The measurement temperature of different methods should be indicated somewhere.

7. 3.4 and 3.5 separated. Please, clarify this fact.

8. The Results and Discussion section is very short and does not include an explanation of the biological application of the liposomal formulation. How it is tripled and what lipids included in the structure of the liposome formulation with vitamins.

9. Please, clarify the choice of the concentrations of vitamins using in the work. In this regard, it is necessary to discuss the detergent effects of compound on the properties of lipid membranes.

10. Please, clarify the toxicity of novel formulation of liposomal vitamin C and vitamin D3.

11. Moreover, my great concern is related to the conclusions. They are too short and schematic. They should be modified. The most important findings of this work should be supported by results and their biological significance should be clearly specified.

Author Response

Authors Response to Reviewer comments on MS ID:  bioengineering-2047251

Reviewer #3

The manuscript is devoted to the study of novel formulation of liposomal vitamin C and vitamin D3. I believe that this manuscript is interesting and should be publishable in this journal; however there are several scientific aspects of this manuscript that I feel the authors must first address.

  1. The abstract must be rewritten and will be contain more detail experimental information.

Response: The authors agree with the reviewer that the abstract required improvement, and the details of experimental information were addressed in the revised manuscript.

  1. line 35-37. Authors indicated on the four different type of liposome. Please, clarify the absence of the information about of giant unilamellar liposomes (GUV).

Response: Giant unilamellar liposomes (GUV) has been added to the introduction of revised manuscript.

  1. line 59-60. Please, explain the abbreviations.

Response: Abbreviations were added to the revised manuscript.

  1. The introduction a very large collection of different information, and refers to many other findings in the literature. A clear message is missing as the paper is very long and poorly structured.

Response: The authors wish to thank the reviewer for this comment. The introduction was updated in the revised manuscript.

  1. Please, clarify the choice of the lipids composition of the novel formulation.

Response:  Bovine and egg-derived lecithin (e.g., egg-yolk, and milk lecithin) can use for preparation of liposome but they have stability issues because of  their high concentration of polyunsaturated fatty acids [1].

1. Le, N.T.T.; Cao, V.D.;  Nguyen,, T.N.Q.; Le, T.T.H.; Tran, T.T.; Thi, T.T.H. Soy Lecithin-Derived Liposomal Delivery Systems: Surface Modification and Current Applications. Int. J. Mol. Sci. 2019, 20(19), 4706-4733.

  1. The measurement temperature of different methods should be indicated somewhere.

Response: The analysis of lipid-based vitamin C and lipid-based vitamin D3 was conducted at room temperature (25 ℃), which was added to the revised manuscript.

  1. 4 and 3.5 separated. Please, clarify this fact.

Response: The analytical methods of quantifying vitamin C and vitamin D are different due to their chemical properties such as solubility since vitamin C is water soluble and vitamin D3 is fat soluble. Therefore, the quantification of the two different compounds was presented separately.

  1. The Results and Discussion section is very short and does not include an explanation of the biological application of the liposomal formulation. How it is tripled and what lipids included in the structure of the liposome formulation with vitamins.

Response: Thank you for the valuable comments. This work is mainly focused on optimization of the lipid-based formula of water soluble and fat-soluble vitamins with simple method and food grade materials. Further investigation may involve the application of liposomal assemblies. The TEM characterization of the liposomes that were formed with bilayer (unilamellar) vesicles, where sunflower lecithin was used in the liposome formulation.

  1. Please, clarify the choice of the concentrations of vitamins using in the work. In this regard, it is necessary to discuss the detergent effects of compound on the properties of lipid membranes.

Response: The concentration of vitamins was selected based on Health Canada’s monograph of natural health products. Liposome assemblies can be achieved by amphiphilic lipids that self-assembles into bilayers based on hydrophobic effects, which contributes to the unique structure-function properties of liposomes as biomembranes and carriers of vitamins [1]. Lipid-based formulation can promote transcellular delivery by temporarily disrupting cellular lipophilic bilayers as well as enhancing paracellular uptake of vitamins [2,3].

  1. Lasic, D.D. Applications of liposomes, in: Handbook of Biological Physics, Elsevier, 1995; pp.491–519.
  2. Wechtersbach, L.; Poklar Ulrih, N.; Cigic, B. Liposomal stabilization of ascorbic acid in model systems and in food matrices. LWT- food sci. technol. 2012, 45(1), 43–49.
  3. Hickey, S.; Roberts, H.J.; Miller, N.J. Pharmacokinetics of oral vitamin C. Nutr. Environ. Med. 2008, 17(3), 169–177.

  1. Please, clarify the toxicity of novel formulation of liposomal vitamin C and vitamin D3.

Response: The simple preparation method of lipid-based formulation uses only food grade materials and processing procedures, which are considered as non-toxic overall.

  1. Moreover, my great concern is related to the conclusions. They are too short and schematic. They should be modified. The most important findings of this work should be supported by results and their biological significance should be clearly specified.

Response: The authors agree with the comment by the reviewer and the conclusion was modified in the revised manuscript. The biological significance was added into the conclusion as a possible research direction in future studies.

In summary, the authors wish to acknowledge reviewer #3 for the insightful and constructive comments. The revised manuscript was comprehensively edited for language, clarity, and syntax throughout to meet the high standards of this journal.

Round 2

Reviewer 1 Report

Minor comments

1. Page 4, line 157 “Potassium sorbate (100.53%)”, please correct

2. Page 4, line 168 “potassium sorbate (100.53%)”, please correct

3. Page 4, line 172 “and blended for 30 min”, the speed is need it.

4. Page 4, line 182 “and blended to obtain liposome”, the speed is need it.

5. Page 6, 3.1. Optimization of formulation, line 260. It is unclear what is the approach followed by the authors in the optimization of formulation. Furthermore, a schematic image for the preparation procedure should be provided.

6. Page 7, Table 2 Content (%) of Sample is 102.24 %, please correct.

7. Page 8, Table 3, the same. The Content (%) of Sample more than 100%, please explain/or correct.

Author Response

Authors’ Response to Reviewer Comments on MS ID: bioengineering-2047251

Reviewer 1 – round 2

Minor comments

  1. Page 4, line 157 “Potassium sorbate (100.53%)”, please correct

Response: Thank you for your comments. The purity of potassium sorbate has been corrected to “99.47±0.5%”.

  1. Page 4, line 168 “potassium sorbate (100.53%)”, please correct

Response: Thank you for your comments. The purity of potassium sorbate has been corrected to “99.47±0.5%”.

  1. Page 4, line 172 “and blended for 30 min”, the speed is need it.

Response: Thank you for your comments. The speed of blending at 3000 rpm has been added.

  1. Page 4, line 182 “and blended to obtain liposome”, the speed is need it.

Response: Thank you for your comments. The speed of blending at 3000 rpm has been added.

  1. Page 6, 3.1. Optimization of formulation, line 260. It is unclear what is the approach followed by the authors in the optimization of formulation. Furthermore, a schematic image for the preparation procedure should be provided.

Response: Thank you for your comments. The optimization of formulation has been revised for indicating key points of optimizing the formulation such as the ratio of water to glycerin and the order of mixing each ingredient, along with addition of Scheme 1 in the revised manuscript.

  1. Page 7, Table 2 Content (%) of Sample is 102.24 %, please correct.

Response: Thank you for your comments. The “Content (%) of Sample” has been corrected to “Recovery value (%)”. There was 5-10% overage of input quantity for vitamins according to the sample product formula to compensate for loss due to degradation of the nutrients during the product’s shelf life and to compensate for the inherent variability of the manufacturing process and product testing.

  1. Page 8, Table 3, the same. The Content (%) of Sample more than 100%, please explain/or correct.

Response:  Thank you for your comments. The “Content (%) of Sample” has been corrected to “Recovery value (%)”. There was 5-10% overage of input quantity for vitamins according to the sample product formula to compensate for loss due to degradation of the nutrients during the product’s shelf life and to compensate for the inherent variability of the manufacturing process and product testing.

In summary, the authors appreciate the insightful and constructive comments provided by Reviewer #1. The revised manuscript was carefully edited throughout for language, clarity, and syntax to meet the high standards of this journal, Bioengineering.

Reviewer 2 Report

The authors addressed most of the comments. Even though the application of liposomal study is not presented, the manuscript is improved with several major modifications to the text. I recommend this articles to publish in its current form.

Author Response

Authors’ Response to Reviewer Comments on MS ID: bioengineering-2047251

Reviewer 2 – round 2

The authors addressed most of the comments. Even though the application of liposomal study is not presented, the manuscript is improved with several major modifications to the text. I recommend this articles to publish in its current form.

Response:  The authors appreciate the constructive comments provided by Reviewer #2. The manuscript was further edited throughout for language, clarity, and syntax to meet the high standards of this journal, Bioengineering.

Reviewer 3 Report

This manuscript is interesting and should be publishable in this journal

Author Response

Authors’ Response to Reviewer Comments on MS ID: bioengineering-2047251

Reviewer 3 – round 2

This manuscript is interesting and should be publishable in this journal

Response:  The authors appreciate the constructive comments provided by Reviewer #3. The manuscript was further edited throughout for language, clarity, and syntax to meet the high standards of this journal, Bioengineering.
